# The Effect of Melatonin Intake on Survival of Patients with Breast Cancer—A Population-Based Registry Study

**DOI:** 10.3390/cancers14235884

**Published:** 2022-11-29

**Authors:** Leda Pistiolis, Djino Khaki, Anikó Kovács, Roger Olofsson Bagge

**Affiliations:** 1Department of Surgery, Institute of Clinical Sciences, Sahlgrenska Academy, University of Gothenburg, 41345 Gothenburg, Sweden; 2Department of Surgery, Sahlgrenska University Hospital, 41345 Gothenburg, Sweden; 3Department of Clinical Pathology, Sahlgrenska University Hospital, 41345 Gothenburg, Sweden; 4Wallenberg Center of Translational Medicine, University of Gothenburg, 41345 Gothenburg, Sweden

**Keywords:** melatonin, breast cancer, survival

## Abstract

**Simple Summary:**

Thus far, studies have shown that the natural hormone melatonin has potential effects on breast cancer development and progression. By integrating data from different Swedish registries, a total of 37,075 patients diagnosed and treated for breast cancer were identified, with 926 of them having been prescribed melatonin. The multivariate analysis showed no significant effect of melatonin on either breast-cancer-specific or on overall survival.

**Abstract:**

Previous research has demonstrated the antitumoral effects of melatonin on breast cancer in both in vitro and in vivo studies. The aim of the present study was to investigate whether melatonin has a favorable effect on the survival of patients diagnosed with early breast cancer. This retrospective registry-based study included all patients diagnosed with breast cancer in Sweden between 2005 and 2015. Data were linked to the Swedish Prescribed Drug Registry and the Swedish Cause of Death Registry. A multivariate Cox regression model, including patient age, tumor size, tumor grade, ER status, HER2 status, nodal status and defined daily doses (DDDs) of melatonin, was used to analyze breast-cancer-specific survival as well as overall survival. Of the 37,075 included patients, 926 (2.5%) were prescribed melatonin, with a median DDD of 30. Melatonin was found to have a protective effect on breast-cancer-specific survival (BCSS) in the univariate analysis (HR: 0.736, 95% CI: 0.548–0.989, *p* = 0.042), but when adjusting for known prognostic factors in the multivariate analysis, this beneficial effect disappeared (HR: 1.037, 95% CI: 0.648–1.659, *p* = 0.879). Melatonin was not proven to have a favorable effect on the survival of patients diagnosed with early breast cancer in this retrospective registry study.

## 1. Introduction

Melatonin was first isolated by Lerner et al. in 1958, from bovine pineal tissue [1]. Shortly after its discovery, it became evident that this small molecule maintained a much larger repertoire than originally thought. Apart from being the primary regulator of circadian rhythm and seasonal breeding, which consist its principal function in vertebrates, it has maintained its ancestral antioxidant capacity. Furthermore, during the course of evolution, it has also acquired anti-inflammatory, antithrombotic, antihypertensive and antitumor effects [2,3,4].

In 1978, the melatonin hypothesis was proposed by Cohen et al. [5], stating that reduced melatonin production, irrespective of the cause, results in a state of “relative hyperestrogenism” which can induce the development of breast cancer. According to Cohen and colleagues, this could be one of the reasons for increased breast cancer incidence in elderly patients, since melatonin production decreases with age. In animal models of breast cancer, melatonin administration suppresses the growth of DMBA-induced mammary tumors [6], reduces the rate of tumor growth [7] and decreases the number of metastases [8]. These effects are mediated by the direct antioxidant and anti-inflammatory effects of melatonin on the tumor microenvironment, as demonstrated by measurements in peripheral blood, and also by a direct anti-angiogenic effect, as depicted by the reduced expression of VEGF-A and its 1 and 2 receptors [8]. The treatment of breast cancer cell lines with melatonin demonstrates reduced viability [9], increased apoptosis [10], inhibition of cell growth [11] and reduced metastatic potential [12]. These actions are mediated by the MAPK, adenylate cyclase, and calmodulin pathways, to name a few [13]. Apoptosis is mediated by the activation of the caspase pathway, TGF-β1 and, probably, p53 [14].

The intricate interplay between melatonin secretion and reduced estrogen biosynthesis has also been investigated, and even though the exact mechanism has not been fully delineated, it is generally agreed that melatonin affects steroid hormone production in three different ways by acting on the hypothalamus (GnRH neurons), the pituitary and the gonads themselves, via the PKA, PKC and MAPK pathways, which are melatonin receptor mediated [15]. Melatonin inhibits estrogen receptor (ER) gene expression by suppressing its transcription [16], and the binding of the ER-estrogen composite to the DNA estrogen response element [17] in MCF-7 (ER positive) breast cancer cell lines. It therefore acts as a selective estrogen receptor modulator (SERM), very much like tamoxifen, the first antihormonal treatment for breast cancer. Furthermore, it decreases aromatase activity in MCF-7 cells and acts as a selective estrogen enzyme modulator (SEEM), such as aromatase inhibitors [17,18,19,20]. Epidemiological studies demonstrate an increased risk for breast cancer in women working night shifts [21,22], a fact attributed to disrupted melatonin production.

A few clinical trials have investigated the effects of melatonin in patients with stage IV cancer, with or without the concomitant administration of chemotherapy [23,24], with promising results. A total of 20 randomized controlled trials have been included in a recent meta-analysis with patients with various types of histologically confirmed cancers, receiving appropriate treatment with or without melatonin supplementation. Tumor response and patient survival were found to be improved and therapy side effects were reduced in the melatonin groups. Trials recruiting solely breast cancer patients have also been conducted, but they mainly concentrated on the effects of melatonin on the side effects of various therapeutic modalities [25,26] or on sleep disturbances as well as depression associated with breast cancer [27,28,29].

The present study aimed to explore the effect of melatonin on the survival of postmenopausal patients, defined as being over 55 years of age and diagnosed and treated for early breast cancer. Studies have shown decreased melatonin production with increasing age [30,31] and, wanting to keep in line with the Cohen hypothesis, we decided to include postmenopausal patients only. Data were retrieved from national Swedish registries. We hypothesized that melatonin would have a positive impact on breast-cancer-specific survival (BCSS), comparing patients who had been prescribed melatonin to those who had not.

## 2. Materials and Methods

### 2.1. Study Design and Patient Characteristics

This retrospective registry-based study of the Swedish population included female patients aged 55 years or more and diagnosed with primary invasive breast cancer between 1 January 2005 and 31 December 2015. A total of 38,120 patients were identified from the Swedish Breast Cancer Registry, and after exclusion of patients with prior breast cancer, bilateral breast cancer, generalized (stage IV) disease at diagnosis or planned for preoperative chemo or endocrine therapy, a total of 37,075 patients were included.

Patient data, including age and menstrual status at diagnosis, and tumor characteristics (i.e., tumor size, Nottingham Histological Grade (NHG), estrogen receptor (ER) status, progesterone receptor (PR) status, Ki-67, human epidermal growth factor receptor 2 (HER2) status and axillary lymph node metastases) were retrieved, including known and established prognostic and predictive factors for breast cancer. The type of breast surgery (mastectomy or breast conservation), axillary surgery (sentinel node, axillary node dissection, both, or axillary sampling) and adjuvant therapy (chemotherapy, radiation therapy, anti-HER2 therapy and endocrine treatment) for each patient were also obtained.

The patient cohort was then linked to the Swedish Prescribed Drug registry to identify those who had been prescribed melatonin, and 945 patients were identified. The defined daily dose (DDD) of melatonin (ATC code: N05CH01) is defined as 2 mg per day according to the WHO. The total DDD for each patient was then calculated as the number of pills prescribed after their breast cancer diagnosis multiplied by the DDD. The overall survival (OS) and breast-cancer-specific survival (BCSS) were calculated using data from the Swedish Cause of Death Registry. The study was approved by the Regional Ethical Review Board of the University of Gothenburg (Dnr: 2019-03481).

### 2.2. Statistical Analysis

For the descriptive data, a standardized statistical analysis was performed with the Mann-Whitney U test for nonparametric continuous variables, presented as the median and interquartile ranges (IQRs). As for the categorical variables, crosstabs analyses were used along with Pearson’s chi-square test, and they are presented in numbers and percentages. A Cox regression analysis was performed both in a univariate and a multivariate setting, presented as a hazard ratio (HR) with a 95% confidence interval (95% CI). The additional variables analyzed in the multivariate analyses for both breast-cancer-specific survival (BCSS) and overall survival (OS) included age, tumor size, grade, axillary lymph node metastasis, Ki-67, ER status, HER2 status and exposure/no exposure to melatonin. For visualization, Kaplan–Meier curves and Cox regression analyses were performed regarding the exposure of melatonin on BCSS and OS. SPSS version 28 (Chicago, IL, USA) was used for the statistical analyses. The statistical significance level was set to *p* < 0.05.

## 3. Results

### 3.1. Patient and Tumor Characteristics

The patients diagnosed with primary invasive breast cancer between 2005 and 2015 were retrospectively identified by the Swedish Breast Cancer Registry and were then linked to the Swedish Prescribed Drug Registry to identify those who had been prescribed melatonin. A total of 37,075 female patients aged 55 and older were included in the study. The treatment group (melatonin) included 926 patients (2.5%) with a median DDD of 60. Overall, the patient and tumor characteristics for both patient groups were similar, except for minor differences in age at diagnosis (68 vs. 67 years, *p* = 0.019), percentage of women diagnosed by screening (53.1% vs. 49.2%, *p* = 0.019) and median tumor size (16 vs. 16 mm, *p* = 0.033) for the non-melatonin and melatonin groups, respectively (Table 1).

### 3.2. Survival Analysis

The 5 year and 10 year BCSS for patients exposed to melatonin was 97.6% and 97.0% compared to the patients not exposed to melatonin at 94.7% and 91.5% (*p* = 0.041) (Figure 1), with a hazard ratio (HR) of 0.736 (95% CI 0.548–0.989), which was statistically significant (*p* = 0.042) (Table 2). However, in a multivariate Cox regression, the only significant independent prognostic factors were age, tumor size, tumor grade, ER status and Ki67 and nodal status, with no benefit of melatonin shown (HR: 1.037, 95% CI 0.648–1.659) (Table 2).

The 5-year and 10-year overall survival (OS) for patients exposed to melatonin was 88.9% and 74.3% compared to 86.2% and 71.0% for the patients not exposed to melatonin (*p* = 0.097) (Figure 2), with an HR of 0.886 (95% CI 0.767–1.022) (Table 2). Subsequent Cox regression analysis for OS yielded the same significant independent prognostic factors as BCSS, with the addition of HER2 status (Table 2). Cox regression analyses of BCSS before and after adjusting variables, are seen in Figure 3 and Figure 4, respectively. Cox regression analysis of OS without adjusting variables is seen in Figure 5.

The effect of a cumulative dose of melatonin intake was studied by dividing the patients into high- and low-melatonin intake groups based on the prescription data, with a cut-off of 190 DDD (corresponding to 3 months of treatment), and the HR for BCSS for the low-melatonin intake group was 1.180 (95% CI: 0.718–1.941) and for the high intake group 0.525 (95% CI: 0.131–2.111).

## 4. Discussion

The present study aimed to investigate the effect of melatonin intake on the survival of patients diagnosed with breast cancer. Three Swedish registries (i.e., the Breast Cancer Registry, the Prescribed Drug Registry and the Cause of Death Registry), all of which are of high coverage nationwide, were used for the data retrieval. In the univariate Cox regression analysis, melatonin was shown to have a protective effect against breast cancer deaths; however, no association was found in the multivariate analysis.

Few trials have been conducted to examine the effect of melatonin on the survival of patients diagnosed with early breast cancer. Lissoni and colleagues published, in 1995, two clinical studies composed of stage IV breast cancer patients. The first study [32] included 14 patients with histologically verified breast cancer, both ER positive and ER negative, with systemic metastases. As the patients were frail or suffered from serious comorbidities, they received tamoxifen as the sole treatment and showed no response or progression after the initial response. The addition of melatonin yielded four patients with a partial response, eight patients with stable disease and two patients with disease progression. The assessment was conducted monthly for the first three months and every three months thereafter. The second study [33] included 40 patients with ER-negative metastatic breast cancer, who were not eligible for chemotherapy due to the presence of comorbidities, and they were treated with tamoxifen alone, for lack of a better treatment. The patients were randomized into groups receiving melatonin or not together with tamoxifen. The patients receiving the combination treatment had a significantly better response rate and increased survival at 1 year.

Surprisingly, despite the promising effect of melatonin exposure in advanced breast cancer patients, the relationship between melatonin and breast cancer has not been further researched in clinical studies, investigating not just for stage IV but also early-stage breast cancer. Preclinical studies provide an abundance of evidence pertaining to the anti-estrogenic effects of melatonin, acting both as a selective estrogen receptor modulator (similar to tamoxifen) [16,17] and as a selective estrogen enzyme modulator (like an aromatase inhibitor) [34]. Research on breast cancer cell lines has demonstrated both the oncostatic and antiproliferative effects of melatonin, especially in the ER-positive subtype [34,35]. The pretreatment of MCF7 breast cancer cells with melatonin increases their sensitivity to ionizing radiation [36], and its co-administration with chemotherapeutic agents potentiates their effects [37]. In a study investigating continuous melatonin release together with tamoxifen on MCF7 cells, a decrease in cellular proliferation and an increase in apoptosis was shown [38]. Last, but not least, melatonin has been found to decrease the epithelial–mesenchymal transition in breast cancer stem cells, thus mitigating their metastatic potential [35].

To the best of our knowledge, this is the first registry study on melatonin intake in early breast cancer patients that has been undertaken, retrieving data from a national breast cancer registry with almost 100% coverage. The results showed no significant association between melatonin and BCSS in the multivariate analysis, even though the association was significant in the univariate analysis. This points towards that patients receiving melatonin is a selected group of patients, with different prognostic factors from the larger cohort. Another finding was that most patients had a limited prescription of melatonin, with a median of only 30 days. We tried to overcome this by dividing the melatonin group into subgroups (high and low) concerning melatonin intake. The decision on the cut-off point was arbitrarily set to 3 months of medication, and, interestingly, there was then a benefit of melatonin in BCSS, with an HR of 0.525, far from being significant, however. Therefore, to fully answer the question of the effect of melatonin, a potential focus should be on patients that received treatment for longer time periods; however, such materials might be hard to identify.

In Sweden, melatonin is prescribed mainly for sleeping disorders, and, consequently, we were able to retrieve this information through the Swedish Prescribed Drug Registry. A caveat is that some patients might have ordered melatonin on the Internet or bought it in another country where it is available over the counter. Unfortunately, we cannot account for this, but if we take into account that the study was retrospective and concerned patients treated 10–15 years ago, when melatonin was not as popular, then the number of unregistered melatonin users should be relatively low. Additional limitations include the relatively small number of patients in the melatonin group compared to the control group and also the relatively low prescription of melatonin in the patients that were prescribed melatonin.

## 5. Conclusions

Taken together, the role of melatonin in breast cancer is still unclear. The present study showed a positive effect of melatonin in the univariate analysis, but when adjusting for the known risk factors, this effect disappeared. We believe, however, that the role of melatonin in breast cancer is worth further investigation. To fully delineate any potential clinical benefits of melatonin on patients with breast cancer, prospective trials are needed.

## Figures and Tables

**Figure 1 cancers-14-05884-f001:**
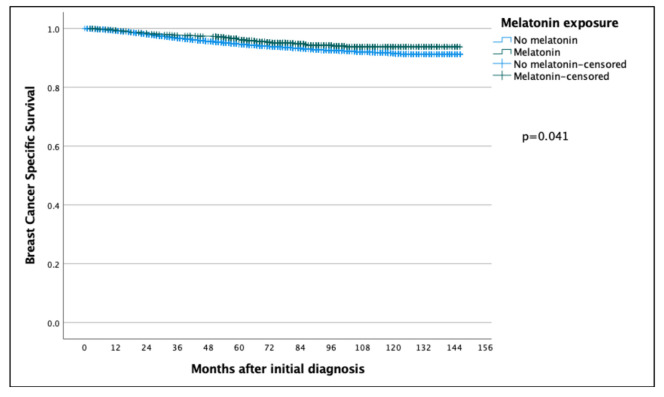
Kaplan–Meier curve comparing breast-cancer-specific survival of breast cancer patients in the melatonin and non-melatonin groups. The difference was found to be statistically significant using the log-rank test, benefitting the melatonin group.

**Figure 2 cancers-14-05884-f002:**
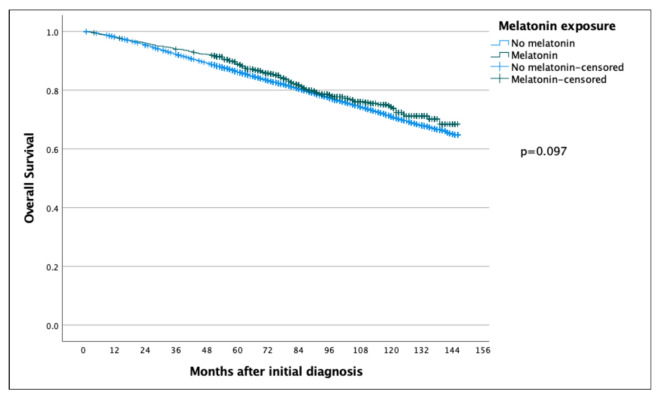
Kaplan–Meier curve comparing overall survival of breast cancer patients in the melatonin and non-melatonin groups. The difference between the two survival curves was not found to be statistically significant when analyzed using the log-rank test.

**Figure 3 cancers-14-05884-f003:**
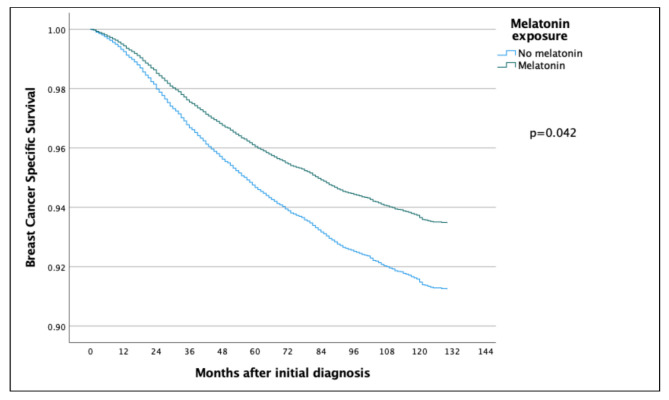
COX regression analyses without adjusting variables, comparing BCSS in the melatonin and non-melatonin groups, showing a statistically significant effect benefitting the melatonin group.

**Figure 4 cancers-14-05884-f004:**
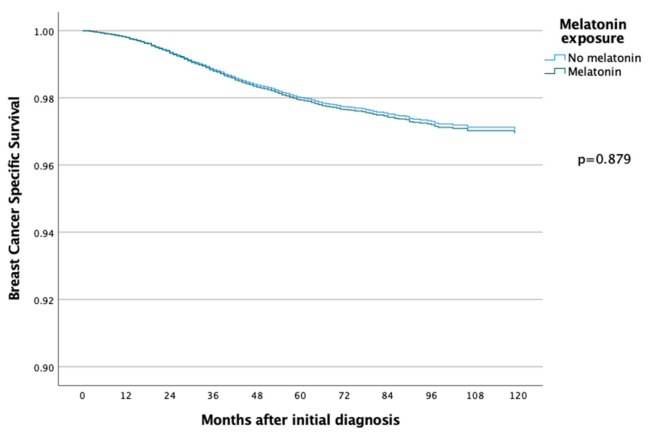
COX regression analyses with adjusting variables, comparing BCSS in the melatonin and non-melatonin groups, and there was then no statistically significant difference between the two groups.

**Figure 5 cancers-14-05884-f005:**
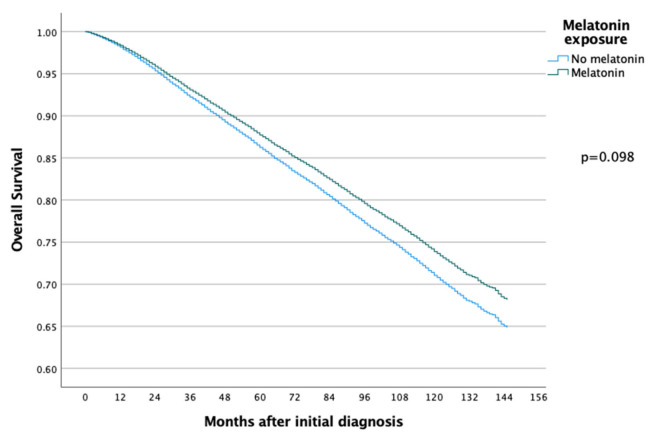
COX regression analyses without adjusting variables, comparing OS in the melatonin and non-melatonin groups, and there was no statistically significant difference between the two groups.

**Table 1 cancers-14-05884-t001:** Overview of patient and tumor characteristics and treatment types in the patient population and as related to melatonin intake.

Variables		All Patients (*n* = 37,075)	No Melatonin (*n* = 36,149, 97.5%)	Melatonin (*n* = 926, 2.5%)	*p*-Value
Age (median, IQR)		68 (62–74)	68 (62–74)	67 (62–73)	0.019
Menstrual status (%) missing: 713	Premenopausal	268 (0.7)	262 (0.7)	6 (0.7)	0.568
	Postmenopausal	34,725 (93.7)	33,864 (95.5)	861 (94.9)	
	Uncertain	1369 (3.7)	1329 (3.7)	40 (4.4)	
Screening diagnosed (%) missing: 142	No	17,343 (46.8)	16,874 (46.9)	469 (50.8)	0.019
	Yes	19,590 (52.8)	19,135 (53.1)	455 (49.2)	
Type of surgery (%) missing: 135	Breast conservation	21,527 (58.1)	20,975 (58.0)	552 (59.6)	0.571
	Mastectomy	15,474 (41.7)	15,101 (41.8)	373 (40.3)	
	No breast surgery	65 (0.2)	64 (0.2)	1 (0.1)	
Axillary surgery (%) missing: 135	No axillary surgery	1171 (3.2)	1142 (3.2)	29 (3.1)	0.645
	Sentinel node biopsy	24,049 (64.9)	23,435 (65.1)	614 (66.3)	
	Axillary node dissection	4873 (13.1)	4766 (13.2)	107 (11.6)	
	SNB + ALND	5920 (16.0)	5770 (16)	150 (16.2)	
	Axillary sampling	927 (2.5)	901 (2.5)	26 (2.8)	
Tumor type (%) missing: 0	Ductal	27,516 (74.2)	26,840 (74.2)	676 (73.0)	0.080
	Lobular	5324 (14.4)	5169 (14.3)	155 (16.7)	
	Other	4235 (11.4)	4140 (11.5)	95 (10.3)	
Tumor size in mm (median, IQR) missing: 668		16 (11–25)	16 (11–25)	16 (11–23)	0.033
NHG (%) missing: 670	Grade I	8031 (21.7)	7819 (22)	212 (23.4)	0.616
	Grade II	18,753 (50.6)	18,296 (51.5)	457 (50.4)	
	Grade III	9621 (26.0)	9383 (26.4)	238 (26.2)	
ER status (%) missing: 864	Positive	31,804 (85.8)	30,994 (87.8)	810 (88.9)	0.311
	Negative	4407 (11.9)	4306 (12.2)	101 (11.1)	
PR status (%) missing: 1076	Positive	26,090 (70.4)	25,429 (72.5)	661 (72.9)	0.783
	Negative	9909 (26.7)	9663 (27.5)	246 (27.1)	
HER2 status (%) missing: 4827	Positive	3750 (10.1)	3655 (11.6)	95 (11.8)	0.898
	Negative	28,498 (76.9)	27,786 (88.4)	712 (88.2)	
Ki67 (median, IQR) missing: 18,379		20 (10–33)	20 (10–33)	19 (10–34)	0.177
T status (%) missing: 668	T1	24,081 (65.0)	23,450 (66.1)	631 (69)	0.076
	T2	11,013 (29.7)	10,752 (30.3)	261 (28.5)	
	T3	1313 (3.5)	1290 (3.6)	23 (2.5)	
N status (%) missing: 1752	N0	24,931 (67.2)	24,299 (70.6)	632 (71.4)	0.090
	N1	7384 (19.9)	7193 (20.9)	191 (21.6)	
	N2	1984 (5.4)	1951 (5.7)	33 (3.7)	
	N3	1024 (2.8)	995 (2.9)	29 (3.3)	
Chemotherapy (%) missing: 6855	Yes	7991 (21.6)	7779 (26.4)	212 (27.3)	0.575
	No	22,229 (60.0)	21,665 (73.6)	564 (72.7)	
Radiation therapy (%) missing: 6827	Yes	20,092 (54.2)	19,603 (66.5)	489 (62.6)	0.022
	No	10,156 (27.4)	9864 (33.5)	292 (37.4)	
Endocrine therapy (%) missing: 6850	Yes	22,952 (61.9)	22,376 (76.0)	576 (74.1)	0.233
	No	7273 (19.6)	7072 (24.0)	201 (25.9)	
HER2 therapy (%) missing: 6863	Yes	2233 (6.0)	2174 (7.4)	59 (7.6)	0.819
	No	27,979 (75.5)	27,262 (92.6)	717 (92.4)	

**Table 2 cancers-14-05884-t002:** Cox Regression analyses with and without adjusting variables for BCSS and OS.

Parameter	Cox Regression—BCSS				Cox Regression–OS			
	Without Adjusting Variables		With Adjusting Variables		Without Adjusting Variables		With Adjusting Variables	
	HR (95%CI)	*p*-Value	HR (95%CI)	*p*-Value	HR (95%CI)	*p*-Value	HR (95%CI)	*p*-Value
Patient age	1.069 (1.065–1.074)	<0.001	1.056 (1.048–1.065)	<0.001	1.104 (1.101–1.106)	<0.001	1.092 (1.087–1.097)	<0.001
Tumor size	1.017 (1.016–1.017)	<0.001	1.017 (1.014–1.020)	<0.001	1.015 (1.014–1.015)	<0.001	1.010 (1.008–1.012)	<0.001
Tumor grade								
NHG 1	Ref		Ref		Ref		Ref	
NHG 2	3.235 (2.691–3.888))	<0.001	2.127 (1.424–3.176)	<0.001	1.420 (1.333–1.513)	<0.001	1.069 (0.946–1.210)	0.285
NHG 3	9.450 (7.887–11.325)	<0.001	3.093 (2.023–4.730)	<0.001	2.243 (2.100–2.395)	<0.001	1.333 (1.145–1.553)	<0.001
ER status								
ER positive	Ref		Ref		Ref		Ref	
ER negative	3.619 (3.318–3.948)	<0.001	2.253 (1.860–2.728)	<0.001	1.738 (1.642–1.838)	<0.001	1.294 (1.147–1.461)	<0.001
HER2 status								
HER2 positive	1.754 (1.573–1.957)	<0.001	0.831 (0.685–1.008)	0.060	1.090 (1.016–1.169)	0.016	0.865 (0.764–0.978)	0.021
Her2 negative	Ref		Ref		Ref		Ref	
Ki67	1.031 (1.029–1.034)	<0.001	1.014 (1.010–1.018)	<0.001	1.015 (1.013–1.016)	<0.001	1.007 (1.005–1.010)	<0.001
N status								
N0	Ref		Ref		Ref		Ref	
N1	3.043 (2.739–3.380)	<0.001	2.391 (1.972–2.901)	<0.001	1.608 (1.523–1.698)	<0.001	1.323 (1.203–1.456)	<0.001
N2	8.995 (8.006–10.105)	<0.001	5.366 (4.332–6.647)	<0.001	3.150 (2.929–3.389)	<0.001	2.123 (1.864–2.417)	<0.001
N3	18.346 (16.234–20.733)	<0.001	10.239 (8.135–12.887)	<0.001	5.551 (5.102–6.039)	<0.001	3.353 (2.873–3.913)	<0.001
Melatonin exposure								
No melatonin	Ref		Ref		Ref		Ref	
Melatonin	0.736 (0.548–0.989)	0.042	1.037 (0.648–1.659)	0.879	0.886 (0.767–1.022)	0.098	0.957 (0.742–1.235)	0.735

## Data Availability

The data presented in this study are available on request from the corresponding author.

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
