# Peer review of "The Effect of Melatonin Intake on Survival of Patients with Breast Cancer—A Population-Based Registry Study"

_cancers, 2022, doi:10.3390/cancers14235884_

Round 1

Reviewer 1 Report

Previous research has demonstrated the antitumoral effects of melatonin on breast cancer in both in vitro and in vivo studies. This study aimed to investigate whether melatonin has a favourable impact on breast cancer survival. It is a retrospective registry-based study. The multivariate analysis showed no significant effect of melatonin on either breast cancer-specific or overall survival. This manuscript reviews 24 articles. The topic of the manuscript is up-to-date, attractive and well-suited for the journal Cancers. The manuscript is well-written and divided into 5 parts. The text is clear and easy to read. The authors used 5 figures and 2 tables. This aid readers' understanding. I suggest checking for small spelling mistakes and grammar errors. Otherwise, I have no major concerns regarding the manuscript.

Author Response

  1. Previous research has demonstrated the antitumoral effects of melatonin on breast cancer in both in vitro and in vivo studies. This study aimed to investigate whether melatonin has a favourable impact on breast cancer survival. It is a retrospective registry-based study. The multivariate analysis showed no significant effect of melatonin on either breast cancer-specific or overall survival. This manuscript reviews 24 articles. The topic of the manuscript is up-to-date, attractive and well-suited for the journal Cancers. The manuscript is well-written and divided into 5 parts. The text is clear and easy to read. The authors used 5 figures and 2 tables. This aid readers' understanding. I suggest checking for small spelling mistakes and grammar errors. Otherwise, I have no major concerns regarding the manuscript.

Response: Thank you very much for your comments. We have taken care to correct spelling and grammar errors.

Reviewer 2 Report

In this study the authors have studied the effect of melatonin on breast cancer patients using Cox regression analysis. They conclude that melatonin does not favorable clinical effect. It is a well written paper and will contribute to the scientific community. I recommend accepting this manuscript with minor review. Please see my comment below
-    Please mention the unit of defined daily dose of melatonin?
-    In the introduction, please elaborate more on the effect of melatonin on estrogen synthesis
-    Figure 1,2: please scale the y-axis for clarity
-    There are few ongoing/completed clinical trials for melatonin in breast cancer. It would be worth mentioning them such as NCT01805089 ; NCT01965522 Also some relevant literature work such as https://doi.org/10.1016/j.tem.2020.08.001; https://doi.org/10.1017/S1462399409000982
-    There are some small typos, please make sure to correct them. E.g. Page 2 line 66

Author Response

  1. In this study the authors have studied the effect of melatonin on breast cancer patients using Cox regression analysis. They conclude that melatonin does not favorable clinical effect. It is a well written paper and will contribute to the scientific community. I recommend accepting this manuscript with minor review.

Response: Thank you very much for your positive comments and for the time and effort you have put into helping us improve the manuscript!

  1. Please mention the unit of defined daily dose of melatonin?

Response: Thank you very much for this comment where we realize that we was not clear in the manuscript. We have now clarified in the 3rd paragraph in the Methods section that one DDD corresponds to 2mg daily according to WHO, and we have therefor also corrected so that we do not report number of pills, but the DDD throughout the manuscript.

  1. In the introduction, please elaborate more on the effect of melatonin on estrogen synthesis

Response: We have now added more background information into the Introduction, linking melatonin and breast cancer. Please see the revised version.

  1. Figure 1,2: please scale the y-axis for clarity

Response: We thank for this comment and we have discussed this thoroughly before submitting. We do think that in general truncation of the y-axis should not be recommended, not to visually over-emphasize potential differences. However, in the adjusted analysis we chose to do so (starting at 90% and 60% respectively), but we felt that the most prudent way would be to keep the full scale (0-100%) for the “primary” Kaplan-Meier curves. If the reviewer and editor still want us to truncate the y-axis, we can of course do this.

  1. There are few ongoing/completed clinical trials for melatonin in breast cancer. It would be worth mentioning them such as NCT01805089 ; NCT01965522 Also some relevant literature work such as https://doi.org/10.1016/j.tem.2020.08.001 andhttps://doi.org/10.1017/S1462399409000982.

Response: We considered the suggested clinical trials. The first one (NCT01805089) examines the effect of melatonin intake on sleep, mood, and hot flashes on breast cancer survivors. The article has now been incorporated into the Introduction. Concerning the second suggested trial (NCT01965522), we were not able to find any published results, except from an ASCO abstract from 2014, and we therefor opted not to reference this since it is not, to our knowledge, formally published even after almost 10 years. Both the other suggested articles have also been incorporated into the Introduction.

  1. There are some small typos, please make sure to correct them. E.g. Page 2 line 66 

Response: We have taken care to correct spelling and grammar errors.

Reviewer 3 Report

In this article submitted by Pistiolis et al., (cancers-1997591), the authors retrospectively investigate the effect of melatonin on the survival of all Swedish patients diagnosed with early breast cancer between 2005 and 2015 (National Breast Cancer Registry). Their results indicate that melatonin has a protective effect on breast cancer specific survival in univariate analysis, however, when adjusting for prognostic factors in a multivariate analysis, the beneficial effect dissapeared, leading the authors to conclude that it is not posible to establish a favourable effect of melatonin on breast cancer patients in this retrospective study.

Obviusly, the number of patients included in the present work is big enough (nearly 40.000 diagnosed with breast cancer and 945 prescribed with melatonin), and therefore, I have to give credit for having compiled the large number of data included in the study.

However, at the end of the discussion section, the authors themselves point out a series of problems that were worrying me all through the manuscript, and that, in my opinion, make the study less relevant: they are not sure whether or not some patients might have taken melatonin without medical prescription and they speculate that probably not many did so since melatonin was not as popular 10-20 years ago (but there is no way to be sure). They also observed that the numbers of patients in the melatonin group is too low in comparison with the number of early breast cancer diagnosed patients, and they also observe that they lack information about the timing of melatonin intake in relation with the timing of diagnosis. Thus the main conclusion of the work is that no safe conclusions can not be drawn about the scheduling of melatonin administration that patients would most benefit from, and I particularly do not understand this last remark, since melatonin, in their results, was not proven to have any favourable effect on the survival of early breat cancer patients.

All this reasons stated above make me take to think the article is not suitable for publication in Cancers, since no conclusive and relevant results have been obtained.

Author Response

  1. In this article submitted by Pistiolis et al., (cancers-1997591), the authors retrospectively investigate the effect of melatonin on the survival of all Swedish patients diagnosed with early breast cancer between 2005 and 2015 (National Breast Cancer Registry). Their results indicate that melatonin has a protective effect on breast cancer specific survival in univariate analysis, however, when adjusting for prognostic factors in a multivariate analysis, the beneficial effect dissapeared, leading the authors to conclude that it is not posible to establish a favourable effect of melatonin on breast cancer patients in this retrospective study. Obviusly, the number of patients included in the present work is big enough (nearly 40.000 diagnosed with breast cancer and 945 prescribed with melatonin), and therefore, I have to give credit for having compiled the large number of data included in the study. However, at the end of the discussion section, the authors themselves point out a series of problems that were worrying me all through the manuscript, and that, in my opinion, make the study less relevant: they are not sure whether or not some patients might have taken melatonin without medical prescription and they speculate that probably not many did so since melatonin was not as popular 10-20 years ago (but there is no way to be sure). They also observed that the numbers of patients in the melatonin group is too low in comparison with the number of early breast cancer diagnosed patients, and they also observe that they lack information about the timing of melatonin intake in relation with the timing of diagnosis. Thus the main conclusion of the work is that no safe conclusions can not be drawn about the scheduling of melatonin administration that patients would most benefit from, and I particularly do not understand this last remark, since melatonin, in their results, was not proven to have any favourable effect on the survival of early breat cancer patients. All this reasons stated above make me take to think the article is not suitable for publication in Cancers, since no conclusive and relevant results have been obtained.

Response: Thank you very much for your comments, which correlate to the limitation as we have pointed out in the Discussion section. Many studies, especially large epidemiological data, have these limitations and we have therefor quite clearly described this in the Discussion. We can only speculate about intake of melatonin from sources other than prescriptions, which also holds true for any other drug that potentially could be bought in another country and being brought to Sweden.

As for the timing of melatonin administration, we realize that we have been unclear in this, both in the Metods and the Discussion. We have in the analysis only included melatonin given after the diagnose of breast cancer, and this has now been clarified in the 3rd paragraph in the Methods section, and we have also removed the comment from the Discussion.

Reviewer 4 Report

Review Report for the Manuscript “The Effect of Melatonin Intake on Survival of Patients with Breast Cancer– a Population Based Registry Study”

Rating the Manuscript

Originality/Novelty: Is the question original and well defined? Do the results provide an advance in current knowledge?

Yes, the authors have analyzed whether melatonin has a favorable effect on the survival of patients diagnosed with early breast cancer.

Significance: Are the results interpreted appropriately? Are they significant? Are all conclusions justified and supported by the results? Are hypotheses and speculations carefully identified as such?

Yes, the results are interpreted well, and the conclusions are justified by the results.

Quality of Presentation: Is the article written in an appropriate way? Are the data and analyses presented appropriately? Are the highest standards for presentation of the results used?

Yes, the article is written well however data representation needs to be improved.

Scientific Soundness: is the study correctly designed and technically sound? Are the analyses performed with the highest technical standards? Are the data robust enough to draw the conclusions? Are the methods, tools, software, and reagents described with sufficient details to allow another researcher to reproduce the results?

Yes, study design, methods and data analysis are explained well in this manuscript. The materials and methods section can be improved.

Interest to the Readers: Are the conclusions interesting for the readership of the Journal? Will the paper attract a wide readership, or be of interest only to a limited number of people? (please see the Aims and Scope of the journal)

Yes, this would be a great article for the researchers in the cancer research field.

Overall Merit: Is there an overall benefit to publishing this work? Does the work provide an advance towards the current knowledge? Do the authors have addressed an important longstanding question with smart experiments?

Yes. This study provides an advancement to the current knowledge. 

English Level: Is the English language appropriate and understandable?

Yes, English language in the manuscript is appropriate and understandable. 

Overall Recommendation: Accept after Minor Revisions

Given below are the comments for each section of the manuscript.

Simple Summary and Abstract

Simple summary and the abstract well summarizes the content of the manuscript.

Introduction

Introduction section can be improved.  

Authors could discuss about the effect of melatonin on breast cancer in detail. They could discuss the facts like, how it supposed to have effect in breast cancer and what is the pathway of ant interactions/reactions taking place.

Line 48: “According to Cohen and colleagues, this could be one of the reasons for increased breast cancer incidence in elderly patients, since melatonin production decreases with age. In animal models of breast cancer, melatonin administration suppresses the growth of DMBA induced mammary tumors, reduces the rate of tumor growth and decreases the number of metastases.”

Here, authors have mentioned how melatonin affects breast cancer in animal models. But what are the reasons/ mechanism/ reaction pathways for these? Is there any information in literature on this?

Line 53: “Treatment of breast cancer cell lines with melatonin demonstrates reduced viability, increased apoptosis, inhibition of cell growth and reduced metastatic potential.”

Again, here authors need to briefly discuss about the mechanisms/reasons for these observations if available in the literature.

Line 60: “A few clinical trials have investigated the effects of melatonin in patients with stage IV cancer, with or without the concomitant administration of chemotherapy, with promising results.”

Authors need to discuss about the results of this clinical studies here. 

Line 64: “The study aimed to explore the effect of melatonin on the survival of post-menopausal patients, defined as being over 55 years of age, diagnosed and treated for early breast cancer.”

Is there a special reason to study post-menopausal patients? If so, authors need to mention the reason for this selection.

Materials and Methods:

Line 76: “Patient data, including age and menstrual status at diagnosis, and tumor characteristics (tumor size, Nottingham Histological Grade (NHG), estrogen receptor (ER) status, progesterone receptor (PR) status, Ki-67, human epidermal growth factor receptor 2 (HER2) status and axillary lymph node metastases), were retrieved.”

What is it important to consider factors like menstrual status at diagnosis, estrogen receptor (ER) status, progesterone receptor (PR) status and Ki-67?

Figures and Tables:

Quality of the figures need to be improved. Figure captions need to be more informative.

Results and Discussion

For me the discussion section looks more like an introduction section. Authors need to focus and discuss about their results in the discussion section. 

References:

Some of the references cited in the article are more than 10 years old. Other than the references with important information authors could try to replace these with new references if possible.

Author Response

Thank you very much for your comments as well as the time and effort you have put into helping us improve the manuscript. 

  1. Introduction section can be improved. Authors could discuss about the effect of melatonin on breast cancer in detail. They could discuss the facts like, how it supposed to have effect in breast cancer and what is the pathway of ant interactions/reactions taking place.

Response: We agree and have now added more details on the mechanisms of action of melatonin on estrogen synthesis and breast cancer in the Introduction, please see the revised version.

  1. Line 48: “According to Cohen and colleagues, this could be one of the reasons for increased breast cancer incidence in elderly patients, since melatonin production decreases with age. In animal models of breast cancer, melatonin administration suppresses the growth of DMBA induced mammary tumors, reduces the rate of tumor growth and decreases the number of metastases.”. Here, authors have mentioned how melatonin affects breast cancer in animal models. But what are the reasons/ mechanism/ reaction pathways for these? Is there any information in literature on this? 

Response: We have now elaborated more on the mechanisms of action of melatonin in the Introduction and also added Reference 8.

  1. Line 53: “Treatment of breast cancer cell lines with melatonin demonstrates reduced viability, increased apoptosis, inhibition of cell growth and reduced metastatic potential.” Again, here authors need to briefly discuss about the mechanisms/reasons for these observations if available in the literature.

Response: We have now outlined the pathways via which the effects are achieved in the Introduction and added References 13 and 14.

  1. Line 60: “A few clinical trials have investigated the effects of melatonin in patients with stage IV cancer, with or without the concomitant administration of chemotherapy, with promising results.” Authors need to discuss about the results of this clinical studies here. 

Response: We have now provided more information on melatonin in clinical trials with cancer patients in general and breast cancer in the Introduction, and we have also added References 25-29. The clinical trials involving breast cancer patients with Stage IV disease remained in the discussion section, where they are analyzed in more detail, since we feel that we needed to stress the fact that even though some trials were conducted as early as 1995, the oncostatic effect of melatonin has not been further investigated in human trials, in spite of the fact that animal experiments and breast cancer cell line research continued.

  1. Line 64: “The study aimed to explore the effect of melatonin on the survival of post-menopausal patients, defined as being over 55 years of age, diagnosed and treated for early breast cancer.” Is there a special reason to study post-menopausal patients? If so, authors need to mention the reason for this selection.

Response: Studies have shown decreased melatonin production with increasing age and, wanting to keep in line with the Cohen hypothesis, we decided to include post-menopausal patients only. We have now added that information in last paragraph of the Introduction.

  1. Line 76: “Patient data, including age and menstrual status at diagnosis, and tumor characteristics (tumor size, Nottingham Histological Grade (NHG), estrogen receptor (ER) status, progesterone receptor (PR) status, Ki-67, human epidermal growth factor receptor 2 (HER2) status and axillary lymph node metastases), were retrieved.”. What is it important to consider factors like menstrual status at diagnosis, estrogen receptor (ER) status, progesterone receptor (PR) status and Ki-67?

Response: These known and established prognostic factors for breast cancers were included to be able to make an adjusted Cox-regression analysis, to verify the hypothesis that melatonin use would be an independent prognostic risk factor. We have now clarified this in the Methods section.

  1. Quality of the figures need to be improved. Figure captions need to be more informative.

Response: We have now updated the figure captions. We have also updated figure 4 and 5 to images with better resolution.

  1. For me the discussion section looks more like an introduction section. Authors need to focus and discuss about their results in the discussion section.

Response: We have now elaborated more on our results and updated the Discussion.

  1. Some of the references cited in the article are more than 10 years old. Other than the references with important information authors could try to replace these with new references if possible.

Response: We have now updated the references, and added the following references:

Amin N, Shafabakhsh R, Reiter RJ, Asemi Z. Melatonin is an appropriate candidate for breast cancer treatment: Based on known molecular mechanisms. J Cell Biochem. 2019 Aug;120(8):12208-15. PubMed PMID: 31041825. Epub 2019/05/02.

Jin Y, Choi YJ, Heo K, Park SJ. Melatonin as an Oncostatic Molecule Based on Its Anti-Aromatase Role in Breast Cancer. Int J Mol Sci. 2021 Jan 4;22(1). PubMed PMID: 33406787. PMCID: PMC7795758. Epub 2021/01/08.

Cipolla-Neto J, Amaral FG, Soares JM, Jr., Gallo CC, Furtado A, Cavaco JE, et al. The Crosstalk between Melatonin and Sex Steroid Hormones. Neuroendocrinology. 2022;112(2):115-29. PubMed PMID: 33774638. Epub 2021/03/29.

Karadas AK, Dilmac S, Aytac G, Tanriover G. Melatonin decreases metastasis, primary tumor growth and angiogenesis in a mice model of breast cancer. Hum Exp Toxicol. 2021 Mar 23:9603271211002883. PubMed PMID: 33754875. Epub 2021/03/24.

Kong X, Gao R, Wang Z, Wang X, Fang Y, Gao J, et al. Melatonin: A Potential Therapeutic Option for Breast Cancer. Trends Endocrinol Metab. 2020 Sep 3. PubMed PMID: 32893084. Epub 2020/09/08.